# Understanding the global subnational migration patterns driven by hydrological intrusion exposure

Renlu Qiao [1,2], Shuo Gao[3], Xiaochang Liu[4], Li Xia [5], Guobin Zhang[2], Xi Meng[6], Zhiyu Liu[7] ✉, Mo Wang[8] ✉, Shiqi Zhou[7] ✉ & Zhiqiang Wu[1,2,9] ✉

Amid the escalating global climatic challenges, hydrological risks significantly influence human settlement patterns, underscoring the imperative for an in-depth comprehension of hydrological change's ramifications on human migration. However, predominant research has been circumscribed to the national level. The study delves into the nonlinear effects of hydrological risks on migration dynamics in 46,776 global subnational units. Meanwhile, leveraging remote sensing, we procured globally consistent metrics of hydrological intrusion exposure, offering a holistic risk assessment encompassing hazard, exposure, and vulnerability dimensions, thus complementing previous work. Here, we show that exposure is the primary migration driver, surpassing socioeconomic factors. Surrounding disparities further intensified exposure's impact. Vulnerable groups, especially the economically disadvantaged and elderly, tend to remain in high-risk areas, with the former predominantly migrating within proximate vicinities. The nonlinear analysis delineates an S-shaped trajectory for hydrological exposure, transitioning from resistance to migration and culminating in entrapment, revealing dependence on settlement resilience and adaptability.

Migratory processes have historically served as pivotal mechanisms enabling individuals to circumnavigate the risks associated with natural disasters, distancing themselves from impoverished and under-developed regions, and eluding conflict and persecution, all in the pursuit of a life characterized by security, equality, and comfort. As delineated in the International Organization for Migration's World Migration Report of 2020, the estimated count of international migrants was approximately 272 million globally as of mid-2019[1]. In light of the escalating ramifications of climate change, the intensity of migration trends is anticipated to undergo an augmentation due to the heightened occurrence of flooding, excessive precipitation, and other severe weather-related phenomena[2–4]. Consequently, the United Nations' Sustainable Development Goals (SDGs), precisely the 11th and 13th objectives, underscore the imperative of mitigating the deleterious impacts ensuing from unequal population migration patterns triggered by the global hydrological risk. This study concentrates on the risks associated with incremental surface hydrological events such as floods, sea level rises, and storm surges, which are defined under the umbrella term hydrological intrusion.

As global warming trends persist, the earth's hydrological cycle gradually intensifies, materializing in various incremental forms, including glacial melting, hurricanes, and extreme precipitation. These

[1]Shanghai Research Institute for Intelligent Autonomous Systems, Tongji University, 1239Siping Road, Shanghai, P.R. China. [2]College of Architecture and Urban Planning, Tongji University, 1239Siping Road, Shanghai, P.R. China. [3]University of Oxford, 11a Mansfield Road, Oxford OX1 3SZ, UK. [4]School of Urban and Regional Science, Shanghai University of Finance and Economics, Shanghai, P.R. China. [5]School of Management, University of Science and Technology of China, Hefei, P.R. China. [6]Faculty of Information Technology, Beijing University of Technology, Beijing, China. [7]College of Design and Innovation, Tongji University, Shanghai 200093, China. [8]College of Architecture and Urban Planning, Guangzhou University, Guangzhou 510006, China. [9]Peng Cheng Laboratory, Shenzhen, China. ✉e-mail: zyliu0917@gmail.com; landwangmo@outlook.com; zhoushiqi1965@outlook.com; wus@tongji.edu.cn

phenomena collectively contribute to the displacement of millions of individuals globally[5,6]. A body of research underscores how sea level rise (SLR), primarily caused by glacial melting, triggers human migration via direct and indirect impacts, such as the destruction of homes and infrastructure, saltwater erosion, and contamination, all of which pose a significant threat to economic development and agricultural production[7–14]. Moreover, the risks of human migration instigated by flooding and precipitation anomalies are also escalating in inland areas[15]. It has been estimated that anomalies in rainfall were accountable for a net displacement of roughly 5 million individuals from 1960 to 2000[16,17]. Similarly, natural disasters have been found to stimulate migration toward areas with improved living conditions[15,18].

Although numerous studies acknowledge the role of climatic factors risk as a catalyst for human movement, it is essential to note that human migration is a complex societal, regulatory mechanism influenced by various factors, including economic circumstances, cultural nuances, political policies, and warfare, among others[19,20]. The heterogeneity inherent in the migration phenomenon, compounded by geographical and societal characteristics, often limits the consensus in research regarding the extent and direction of the impact of hydrological intrusion on migration. Intriguingly, a segment of the literature suggests that hydrological intrusion may not decisively influence migration patterns under specific circumstances[21,22]. People's perceptions of migration vary widely, and in some cases, individuals opt to transform their living conditions to adapt to environmental changes rather than moving[23,24]. In addition, the scarcity of resources, such as reduced freshwater availability, crop failure, and land degradation, brought about by hydrological changes often acts as a major hindrance, trapping inhabitants in areas prone to erosion, particularly in economically disadvantaged regions[25–27].

Considering these multifaceted complexities, the Intergovernmental Panel on Climate Change (IPCC) and the United Nations Office for Disaster Risk Reduction (UNDRR) have proffered a holistic definition of disaster risk that amalgamates three pivotal elements: hazard, exposure, and vulnerability[28,29]. Hazard refers to potential natural events that can cause harm, such as rainstorms or floods. Exposure denotes the presence of people in areas susceptible to hazards, a variable that can shift due to urbanization or demographic changes. Lastly, vulnerability encapsulates the predisposition of these exposed elements to suffer adverse effects when a hazard occurs, a factor heavily influenced by socioeconomic conditions. Together, these elements provide a holistic framework for understanding and addressing the complexities of climate-related risks.

Previous studies have mainly focused on hazard-related assessments, which divide roughly three types of quantified methods: the degree of deviations and fluctuations in extreme precipitation levels, the severity and number of severe incidents like floods and landslides, and the area of floodplains[3,8,21,30–32]. However, some floodplains and high-rainfall areas possess abundant water resources and fertile lands, which will attract more agriculture and fishery practitioners[8,33]. Furthermore, although several international institutions have provided large disaster data backplanes, there always exist some differences in statistical caliber because of the original disaster data from different sources[34], especially some wealthier economies which continually record higher disaster losses than the less-developed areas[35]. In addition, some studies attempted to determine the exposure degree of hydrological intrusion through population statistics in the low-elevation coastal zone (LECZ) or perennial floodplains but presented a vast divergiant[8,36–39]. This is because the distribution of residents is heterogeneous. Notably, areas with high population density may face amplified risks, which means that statistical precipitation levels and floodplains may misestimate the extent of damage to citizens.

On the other hand, with the progressive enrichment of global bilateral migration datasets, numerous studies have delved into the influence of disaster risk on human migration flows at the country

level[3,8,32]. However, climate change-induced migration typically involves relatively short distances, signifying that the majority of affected individuals opt to relocate within their national borders rather than relocating internationally[3,19,40]. Besides, the drivers of hydrological risks may exhibit resilience, with migration effects materializing only when the system pressure surpasses a particular threshold[41,42]. Despite this, most studies have relied on linear models, which only delineate approximate impact trends and may fail to capture this nuanced dynamic[3].

To address these issues, the study computes population-weighted hydrological intrusion, which leverages dynamic monitoring of global surface water facilitated by remote sensing technology, thereby enabling a more accurate characterization of hydrological intrusion exposure (HIE). Additionally, the study amalgamates natural population growth rates with population distribution data to estimate the migration rate (Mig_R), deducing migration patterns at subnational scales[7,43,44]. Building upon this foundation, the present study delves into migration patterns at a subnational level by employing interpretable nonlinear ensemble learning models[45]. To the best of our knowledge, this is the first observational study to explore global subnational-level population migration triggered by hydrological intrusion risk.

## Results

### Uneven distribution of subnational population migration

To elucidate a more comprehensive and inclusive pattern of population migration, the study encompasses a vast scope, encompassing 249 countries and regions, comprising a total of 46,776 subnational administrative units predominantly at the municipal or county level.

To ensure the robustness of the subnational migration estimation method, the study aggregates the subnational migration values at the country level. It calculates the coefficient of determination (R-square) between the estimates and the country-level migration estimates that are derived from the World Population Prospects 2022 (WPP22) (as illustrated in Fig. 1b). The R-square stands impressively at 0.89, providing compelling evidence for the validity and reliability of the subnational migration estimation method. However, it is crucial to acknowledge that the study cannot discern specific characteristics of population flow, such as origin-destination bilateral patterns or the long-short temporal aspects of migration, owing to the utilization of yearly stock data for migration calculations.

The presented Fig. 1a illustrates significant disparities in population migration at the subnational level. Oceania emerges as a predominant departure for out-migration, while Europe and America are conspicuously discerned as primary magnets for incoming migrants. The Getis-Ord Gi* analysis, as depicted in Fig. 1c, reveals a pervasive out-migration trend encompassing regions such as Middle Africa, Central Asia, and Southern Europe. Yet, the clustering effect associated with immigration remains relatively muted, with confidence indices predominantly hovering around the 90% mark.

The average Mig_R value across global regions stands at −0.59%, signifying that a large portion of the world's population is experiencing out-migration. Although the scale of Mig_R appears minor, its implications are profound, especially when viewed against the backdrop of a global population of 8 billion. The World Migration Report 2022 underscores the presence of 281 million international migrants. This highlights the fact that, due to the cumulative effects of small proportional changes over time and their intensified impacts in densely populated areas, even slight variations can exert significant long-term and broad effects on migration patterns worldwide.

### HIE enhances population emigration

Migration represents an outcome influenced by many diverse drivers, encompassing a complex, adaptable, and endogenous nature. Given

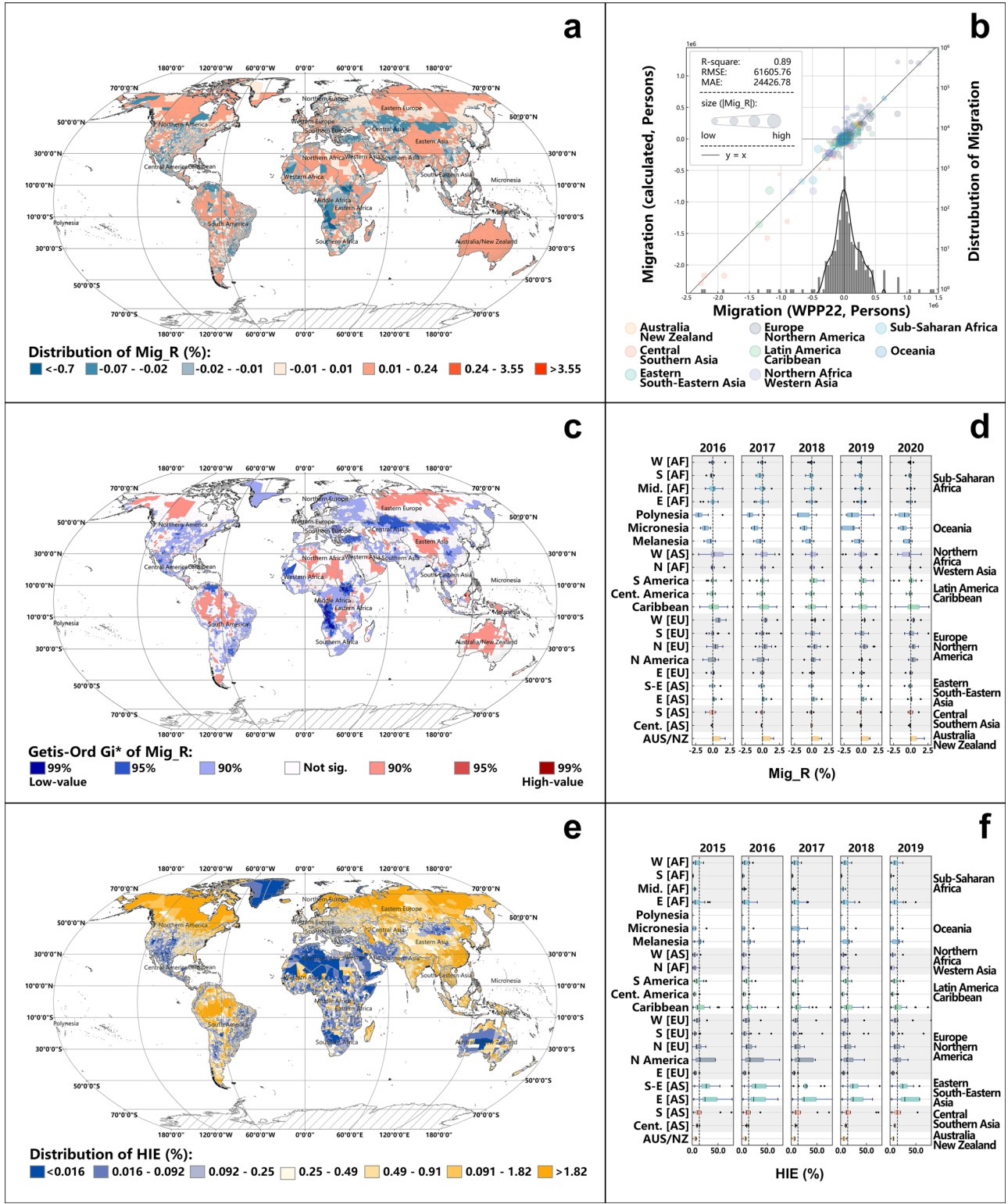

**Fig. 1 | Regional estimates of Mig_R and HIE. a** The geographical distribution of Mig_R, where positive values represent population migration in and negative values represent migration out. **b** The validity of methods for estimating subnational population migration. **c** The hot spot analysis of Mig_R. **d** The distributional of Mig_R. **e** The geographical distribution of HIE. **f** The distributional of HIE.

these intricacies, the study employs an interpretable machine ensemble learning framework comprising a Light Gradient Boosting Machine (LightGBM)[46] and Shapley Additive exPlanations (SHAP)[47] to construct empirical nonlinear models that establish associations between HIE and Mig_R. Moreover, the driver factors underlying migration are inherently multivariate. However, quantifying certain aspects, such as religion, society, and humanities, on a global scale presents considerable challenges. To address this, the investigation adopts a country-year fixed effect model to mitigate the endogeneity issues arising from the above potential omitted variables. The Methods section

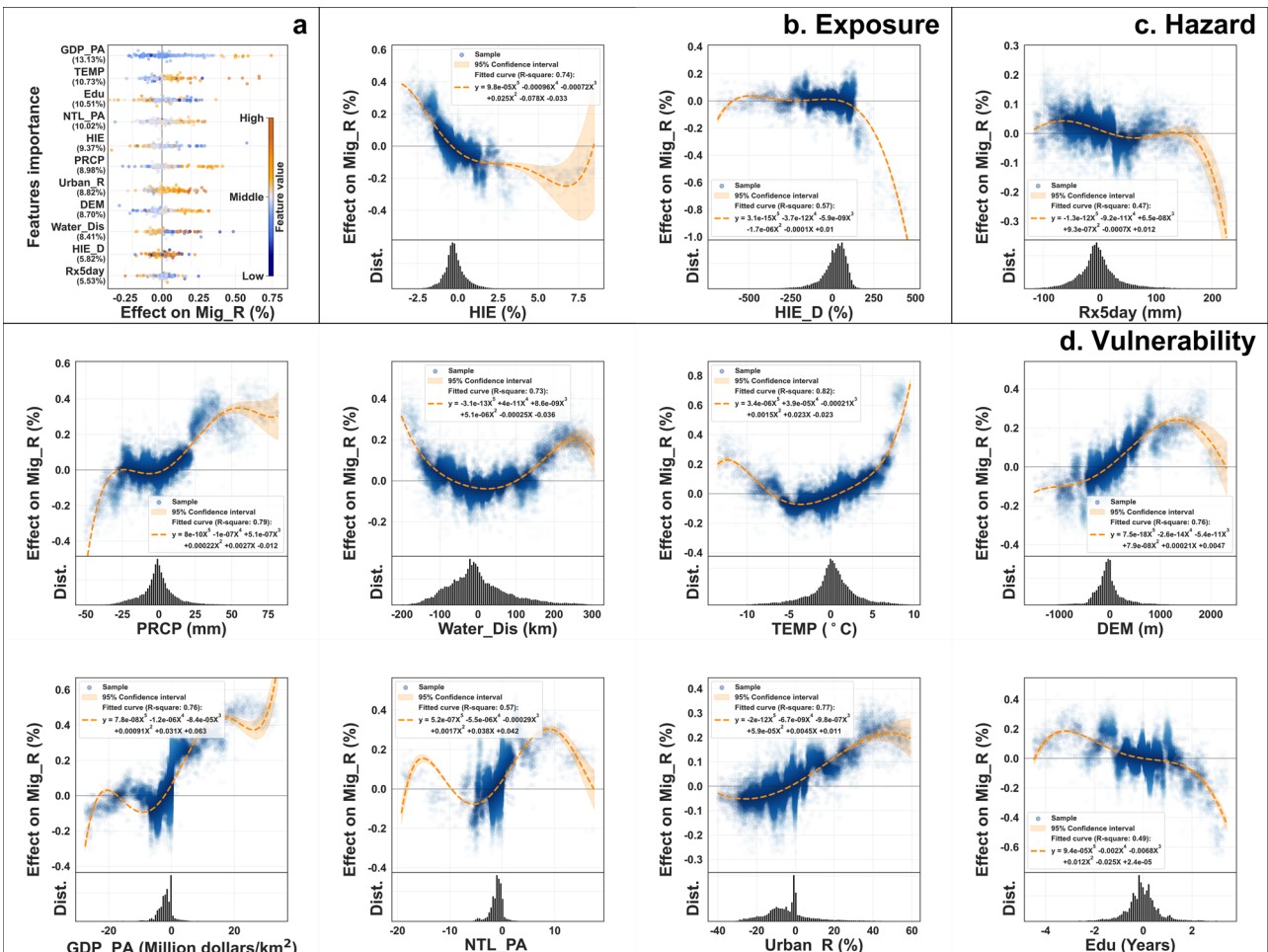

**Fig. 2 | The effect of per feature on Mig_R. a** Ranking of each feature's effect (SHAP values) on Mig_R. **b** Exposure features: HIE and divergence of HIE with neighboring unities (HIE_D). **c** Hazard: Maximum 5-day rainfall (Rx5day). **d** Vulnerability: Average distance from water (Water_Dis), average annual temperature (TEMP), average elevation (DEM), per area GDP (GDP_PA), per area nighttime light (NTL_PA), urbanization rate (Urban_R), and average years of education of the population (Edu).

Noted: in these Subfigures **b**–**d**, the X-axis shows the sample feature value after within-group deviation, and the Y-axis shows the SHAP value of the sample feature, i.e., the local effect on Mig. In Subfigure **a**, the relative importance of each feature is expressed as the ratio of each feature's mean absolute SHAP value to their total value.

thoroughly describes the detailed indicator selection and modeling process.

The results of the global model validated the conclusions reported by others and reaffirmed that increasing hydrological risks lead to regional population displacement. Furthermore, the results also revealed a subtle trend: the migration effect diminishes as the risk exposure increases (as illustrated in Fig. 2b). Specifically, when the HIE falls below 6.74%, an increase in HIE by one standard deviation (1.51%) corresponds to an average increase in the population out-migration rate of 0.09% (with a 95% confidence interval ranging from −0.11 to −0.06%, according to the regression curve). Notably, a higher HIE (>6.74%) appears not favoring outward migration, as is reflected clearly by the inflection point in the regression curve. This is plausibly due to hydrological risks which resulted in regional resource scarcity, which in turn limited the ability of the inhabitants to migrate.

This study sheds light on an unexplored phenomenon. Upon comparing the extent of exposure to hydrological risks in a region with that of its neighbors, one may detect the difference (which we term HIE_D) which significantly amplifies the likelihood of people moving away. Different from HIE, HIE_D reveals an enhanced impact (as shown in Fig. 2b), indicating that certain groups of individuals become more sensitive to HIE when there exists other options and alternatives. In

more tangible terms, as the disparity in HIE exceeds 0%, an increase in HIE by one standard deviation (127.42%) corresponds to a rise in Mig_R by 0.34% (95% CI −0.36 to −0.33). Simply put, when people perceive their area to be at a higher risk compared to that in neighboring regions, the inclination to migrate intensifies.

In addition, socioeconomic variables often exhibit greater relative importance in modeling results. However, a comprehensive analysis reveals that a change in GDP_PA and Edu by one standard deviation corresponds to average changes in Mig_R of 0.05% (95% CI 0.04% ~ 0.06%) and 0.06% (95% CI 0.05 to 0.007%) in the global model, respectively. In contrast, for HIE_D, the average change in Mig_R is 0.16% (95% CI −0.17 to −0.15%), which significantly exceeds the influence of social or economic features. Specifically, the emigration effect caused by HIE relatively slowly increases, and this increase stagnates when the HIE reaches a certain threshold. However, the economic dimension presents a different mechanism of influence altogether. We can notice that the GDP_PA impact shows an explosive boost around the value of 0 and quickly remains flat. This trend suggests that when the region's economic level exceeds the domestic average, it creates a strong attraction for the population, but this attraction does not strengthen as the economy rises further. This mechanism of leapfrogging leads to an overestimation of the economic variables by the statistical result of relative importance.

## The potential effect of income

The mechanism of the HIE effect is intricate and multifaceted, implying that socioeconomic determinants can modulate the scale and direction of HIE impacts through direct or indirect mechanisms[48,49]. To investigate these interactions further, we partition the subnational units into three distinct clusters: low, middle, and high-income groups from the gross national income (GNI) of the country of affiliation computations as delineated by the World Bank standard. In Fig. 3b, the subsequent implementation of the Kolmogorov−Smirnov test (KS) shows evidence that the migration patterns across these three groups exhibit a statistically significant disparity (the $p$ values are all less than 0.05).

As depicted in Fig. 3c–e, the influence of hydrological intrusion on migration patterns exhibits pronounced disparities across different income groups. In the context of the high-income group, an escalation in HIE triggers a phase of relatively minor negative fluctuations in Mig_R. This underscores the robust resilience of high-income residents to hydrological risks. Notably, during the stage of intense exposure (when HIE surpasses 2.40%), the population exhibits a trend toward inward migration. The units in this stage are primarily coastal areas, characterized by abundant economic resources (with GDP_PA exceeding the standard deviation by 0.30 compared to other stages), thus attracting the population. In addition, the result of HIE_D indicates that migration due to neighborhood differences is more pronounced than HIE. Transportation infrastructure in developed regions enhances urban connectivity, making cross-city living possible and allowing residents to choose more liveable environments[50].

As for the middle-income group, the effects of HIE is similar to that of the aforementioned global modeling result, albeit with a lower suppression threshold (as illustrated in Fig. 3d). As HIE falls below 5.00%, an increase in HIE by one standard deviation corresponds to an average decrease in Mig_R of 0.09% (95% CI −0.09% to −0.08%). Compared to the other two groups, the middle-income group appears insensitive to HIE_D. This could be attributed to the fact that, unlike the low-income group, the middle-income group has certain resources to relocate readily. Unlike the high-income group, the middle-income group may not have that much resources to weather through more vulnerability, and relocation is then a necessary option.

In the case of the low-income group, the impact of HIE exhibits similar oscillations to those observed in the high-income group (as illustrated in Fig. 3e). When considered in conjunction with the trend in HIE_D impacts, this may be indicative of migration constraints due to resource scarcity, as opposed to a risk resistance mechanism of high-income groups. In the low-income group, HIE_D emerges as the dominant variable, with its relative importance ranked fourth at 10.52%, significantly surpassing that of the other groups. When the local HIE exceeds that of the surrounding area, an increase in HIE_D by one standard deviation (168.14%) corresponds to an average decrease in Mig_R of 0.47% (95% CI −0.49% to −0.45%). The interplay of HIE and HIE_D effects suggests that the low-income group is vulnerable to the effect of hydrological intrusion but is constrained by resource limitations, with neighborhood migration emerging as its predominant mode.

## Population heterogeneity of hydrological intrusion effect

Populations at varying life stages and sexes may exhibit distinct interactions with the hydrological intrusion. To elucidate the mechanisms underlying these differences, the study stratified the population into three age groups: minor (0−20 years), adult (20−65 years), elder (over 65 years), male, and female, and computed the Mig_R for each group. The computational process is explicitly described in the Subnational migration subsection of the Methods section. Figure 4 presents the modeling results, highlighting the differential impacts of hydrological intrusion across these population segments.

In age demographics, the median Mig_R values for the minor, adult, and senior groups are −1.41, −0.71, and 2.23%, respectively. This suggests that younger age groups in most units predominantly exhibit an outflow trend toward a few areas, while the senior age groups display the inverse pattern. This phenomenon can be attributed to labor mobility, where regional economic disparities and employment opportunities incentivize younger individuals to migrate[51,52]. Conversely, the burden of taxes and post-migration lifestyle pressures often compel migrants to return to their place of origin as they age[53]. Figure 4a–c further underscores the differential vulnerability characteristics among the three age groups. Both the minor and senior groups exhibit a clear trend of diminishing margins, particularly for the seniors, whose impact oscillates around the zero mark as HIE exceeds 0%. Coupled with the observation that HIE_D in the elder group hovers around zero, it can be inferred that the elderly's resilience to environmental change is fragile, trapping them in areas of high hydrological intrusion.

When it comes to sex disparities, the findings of this study show that environmental factors affect both sexes in similar ways, but the effects are more noticeable in men. Specifically, the significance of HIE and HIE_D in influencing decisions is marginally higher for men than women, at 14.82 and 13.04%, respectively. This suggests that men might be more responsive to negative changes in their environment, and migrate, particularly when migration often leads to improved earnings. On the other hand, for women, the link between migration and earnings appears not as strong, and they are therefore less receptive to relocate[54].

## Discussion

This study shed light on the role of regional hydrological risk in subnational population migration. The investigation estimates subnational Mig_R by leveraging inter-annual variations in population distribution and utilizing the population-weighted remote sensing hydrological change to quantify the degree of HIE per unit, thus complementing the missing exposure assessment in previous risk studies. In the nonlinear modeling approach, the study has controlled for the country-year effect and incorporated a comprehensive range of social, economic, geographic, and climate variables. The model's output reveals that HIE and HIE_D substantially influence regional population out-migration, underscoring the significant role of local hydrological risks and neighborhood differences in shaping migration patterns. However, our findings also highlight that hydrological risks do not contribute uniformly to the migration effect experienced by different countries and population groups.

From the results of the global and subgroup models, one can see that the impact of HIE and HIE_D on migration rates often registers below 10%−indicating a seemingly minor role. Similarly, some reported studies have also shown that climatic conditions are weak predictors of migration[20,55]. Nonetheless, one does not dismiss the significance of hydrological factors on population movements as inconsequential. Particularly notable is the role of HIE_D: when the HID-value exceeds the mean value, a 1.35% decrease in Mig_R and represented the most significant reduction attributed to all other parameters examined. The caveat here is the small number of samples that exceed the mean, which, in turn, appear to marginalize the overall importance of HIE_D in the broader analysis. This finding underscores the resilience of a population group to risks, suggesting that while hydrological variables may not always be the primary drivers, their impact, especially in specific contexts like that of HIE_D-value exceeding the mean value, can be substantial.

Among the various identified drivers of migration, the economic condition of a region stands out at a relative importance of 13.13% in driving population shifts. However, the influence of economic condition on Mig_R predominantly hovers around the national average, with a Mig_R improvement of 0.40% within one standard deviation in the

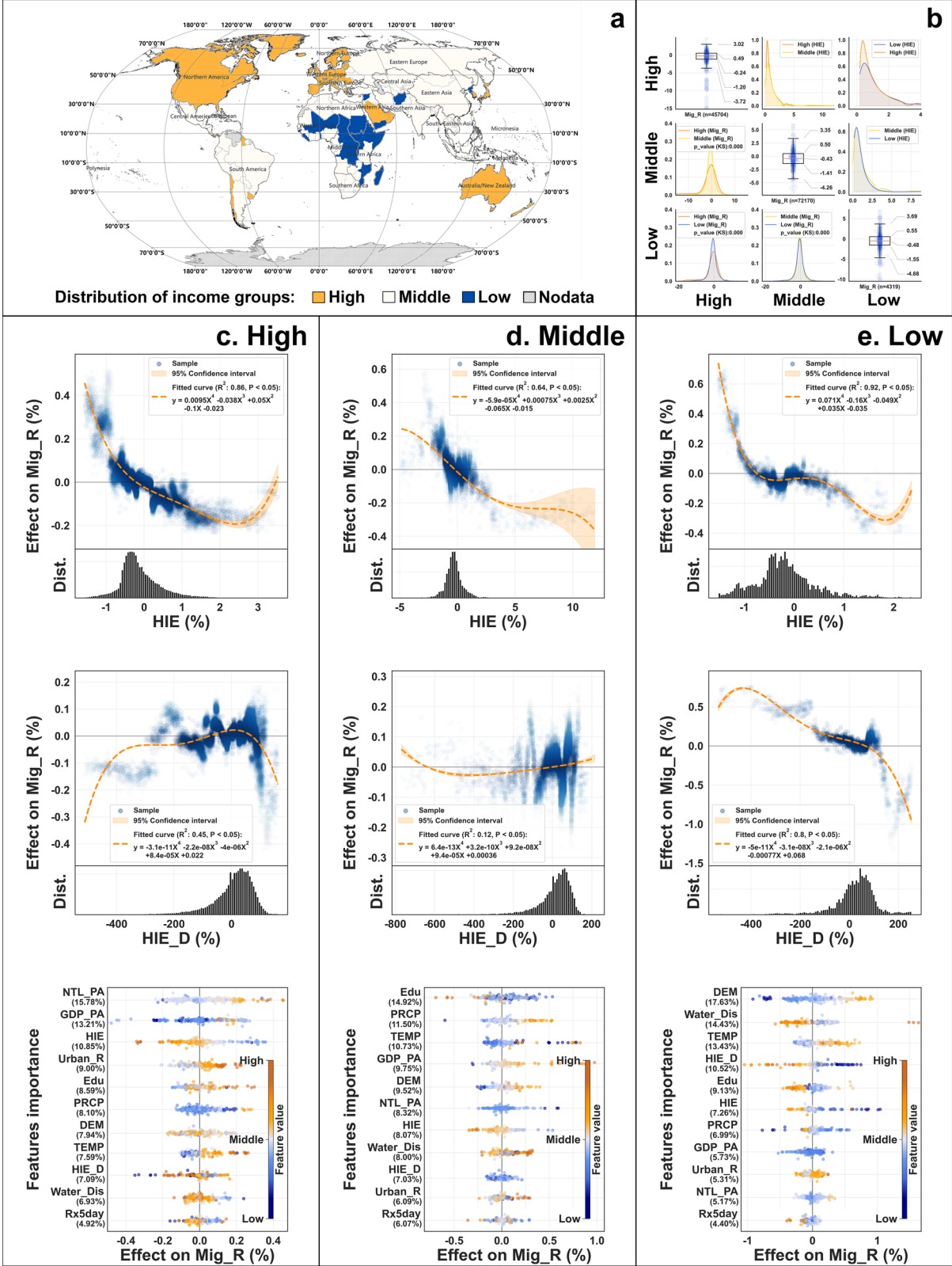

**Fig. 3 | The effect of HIE among income groups. a** The geographical distribution of high, middle, and low-income groups. **b** The distributional difference of Mig_R and HIE. **c**–**e** the SHAP values of features in high, middle, and low-income groups.

economy (6.75 million dollars per square kilometer) above and below the mean. This suggests that while immediate economic enhancements may effectively curb population decline, achieving higher than the average national economy does not necessarily hold the residents more firmly to the home region. In contrast, mitigating hydrological risks emerges as a more viable, long-term strategy to deter out-migration.

Earlier reported studies might have overlooked this nuance, and might have simplistically mapped environmental factors to migration in a linear fashion. The intervention policy might then be less than

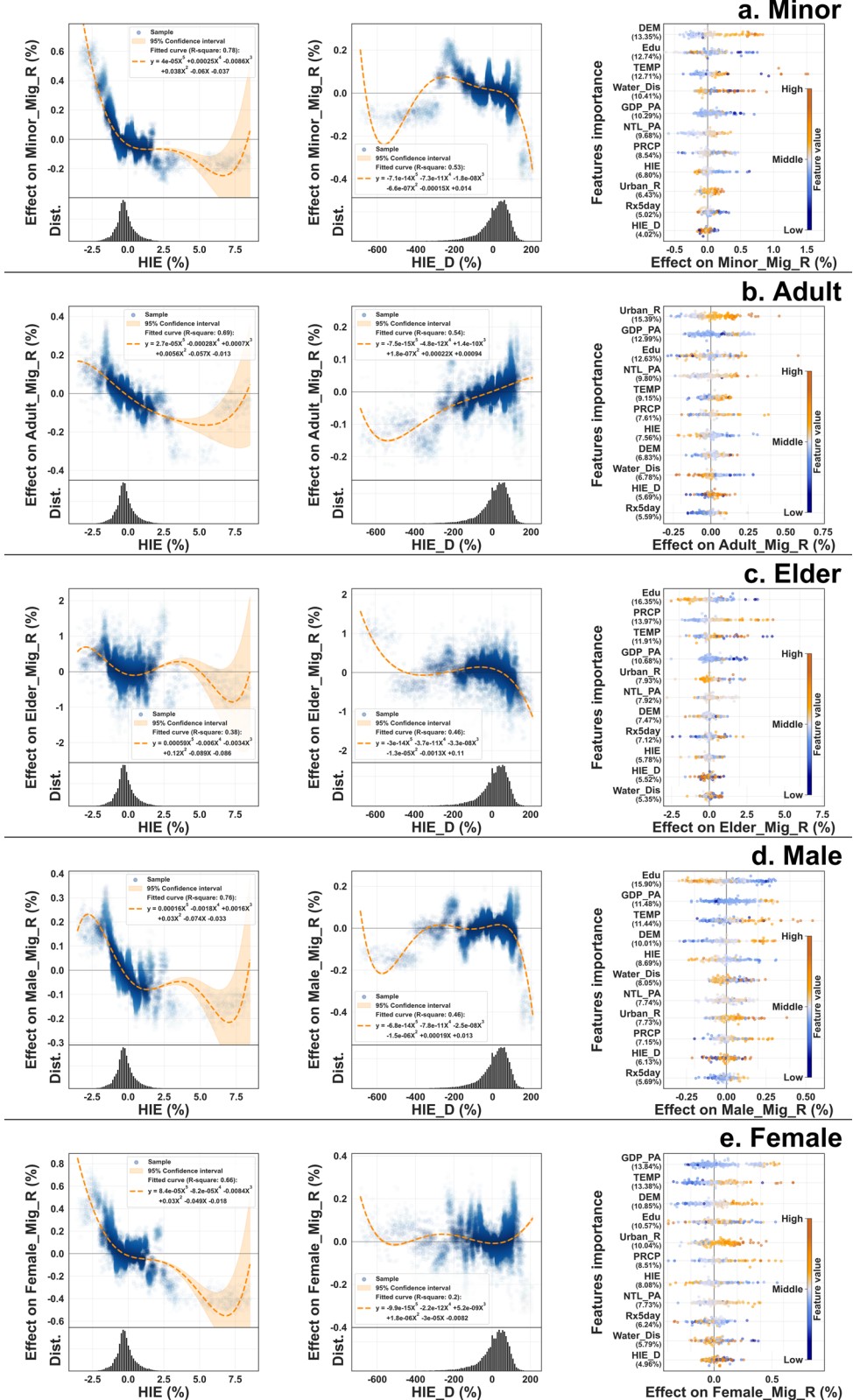

**Fig. 4 | The effect of HIE among population groups. a–e** The SHAP values of features in minor, adult, elder, male, and female groups.

optimal. Based on the findings of this study, the writers advocate for a refined perspective, recognizing that while economic development is crucial, environmental risk mitigation efforts, especially mitigating the differences and gaps with neighboring regions, are positive long-term measures that could be used to manage population displacement more effectively.

Furthermore, securing globally consistent and precise observations for cataloging disaster events remains a formidable challenge, leading to pronounced biases in the global-scale migration model predicated on hazard variables. Our proposed methodology for estimating HIE, underpinned by remotely sensed inversion, boasts spatial and temporal consistency. It accentuates the ramifications of

hydrologic erosion on inhabitants and may position it as a paradigm for ensuing research endeavors on population displacement.

Our study reveals that among different demographic groups—minor, adult, elder, female, and male—the intensity and thresholds of migration impacts due to hydrological risks vary significantly. However, there is a strong positive correlation between increasing environmental instability and the propensity for residents to relocate. This indicates that migration trend in response to hydrological change is influenced by a constellation of potential factors, chief among them being the vulnerability of inhabitants and their adaptive capacity within regions confronted by escalating or heightened hydrological risks[56,57]. The degree of vulnerability is contingent upon intricate interplays of economic, political, and sociocultural dynamics prevalent in the area, alongside the inherent demographic traits[58]. Furthermore, severe environmental disruptions can hinder economic progress, thereby narrowing the range of choices and capabilities available to the affected communities in mitigating such challenges[59]. Nonetheless, populations with a robust adaptive capacity are more adept at navigating and adjusting to hydrological risks without resorting to large-scale migration[60].

Economic development is pivotal in orchestrating the influence dynamics of hydrological encroachment on cross-border migration patterns[61]. This role is clearly elucidated in this study and is discernible at the subnational level. Notably, a high-income region portrays remarkable resilience, wherein even locales fraught with heightened hydrological jeopardy (HIE ≥2.40%) exert a pull on inhabitants due to their strong economic status. In contrast, regions characterized by low-income strata show a tendency to stay put in the face of risks, wherein the parameter denoting HIE_D (relative importance at 10.52%) ascends to prominence as the most impactful parameter with respect to HIE and hazardous conditions. This conveys a double whammy confronting economically disadvantaged groups, i.e., increased vulnerability to calamities and dwindling resources to escape from them. As a result, migration, if at all is restricted to adjoining areas.

This observation is reflective of those certain demographic segments, notably older adults and women, who are exposed to heightened vulnerability, and are ensnared in regions with extensive hydrological risk. An added observation is that the elderly demographic manifests pronounced tendencies towards a stable environment and the desire to move away from metropolitan areas. Such trends can be ascribed to an amalgamation of reasons, such as the symbolic falling leaves returning to the roots (getting home) cultural concept and a passive stance due to aging and weaker physical conditions[62]. On the other hand, the characteristics of the population group play a vital role in shaping migration patterns. Objective physical factors such as employment opportunity and subjective perceptions of risk, cultural norms, and religious beliefs can significantly moderate decision-making processes, either facilitating or resisting migration[61,62].

Migration may be viewed as both a response and pre-emptive measure to climate change, and constitute certain potential strategies for resettlement[63]. This study reveals that the impact of environmental changes on migration is nonlinear, and is especially evident in the exposure parameter HIE_D and the hazard parameter Rx5day. Both of these parameters demonstrate the resilience of the population group against environmental stresses, with out-migration occurring only after the critical threshold is exceeded (−37.15% for HIE_D and 135.66 mm for Rx5day). This may be the direct consequence of the population group being overwhelmed, ceased to accept, and unable to adapt to emerging pressures[41,64]. The identification of these thresholds is crucial, as the threshold signifies the limits of adaptability and resilience within the population group, beyond which significant population movements would be observed.

Moreover, the study observed a plateau in migration as these risks intensified further (less than 6.74% based on the overall model). The trend delineates an S-shaped trajectory, transitioning from resistance to migration to adaptation, ultimately leading to entrapment. While the heterogeneity models indicate that the thresholds for migration and entrapment vary marginally, the overarching patterns remain consistent. This consistency underscores the robustness of research findings about the influence of hydrological intrusion on migration. It's noteworthy to mention that most studies within the migration domain predominantly rely on linear models[65]. While attempting to capture the intricate nonlinear local nuances of the relationship, such models may inadvertently omit pivotal information.

The work renders research gains added relevance in light of recent extreme hydrological events such as the Libyan floods. The extreme precipitation in Libya swiftly overwhelmed local resilience, stranding tens of thousands in flooded areas. There is an imperative need for enhanced early warning systems and community awareness programs. These should be designed to not only alert communities of impending dangers but also educate and empower them on effective response strategies. Strengthening infrastructure resilience through the reinforcement of flood defenses and the development of appropriate sustainable water management solutions are also crucial. This catastrophic event is a stark reminder of the increased hydrological risks that vulnerable communities increasingly face. It also highlights the urgent need for comprehensive, and adaptive strategies in addition to immediate risk mitigation, including long-term preparedness for future hydrological intrusion events.

To mitigate the impact of natural disasters, it is crucial to enhance infrastructure in economically disadvantaged areas. This involves strengthening housing structures, constructing flood defenses, and upgrading drainage systems to offer better protection to residents during such events, thereby diminishing the necessity for relocation. Furthermore, the provision of comprehensive social support and services is essential. This includes access to healthcare, services catering to the elderly, and economic assistance tailored for individuals with limited physical abilities or those belonging to other vulnerable groups. Implementing these measures aims to alleviate the burden of natural risks, enhancing the capability of individuals and communities to manage and respond to natural disasters effectively.

Furthermore, the analysis raises pertinent critiques regarding the pervasive reliance on linear models in preceding migration research. The inherently nonlinear trajectories of environmental influences on migration, punctuated by discernible tipping and inflection points, underscore the urgency of proactive policy measures. A passive stance, predicated on the emergence of overt migration patterns, might culminate in missed opportunities for timely and efficacious interventions. As climate change intensifies hydrological and associated environmental threats, a profound understanding of these intricacies becomes indispensable for policymakers, urban planners, and communities.

This research recognizes several inherent limitations that merit consideration. While the primary focus has been on micro-migration patterns at the subnational level, challenges in data acquisition have impeded the comprehensive construction of bilateral migration relationships. Cellular phone data might serve as a pivotal resource, offering enhanced insights into domestic migration trajectories. Furthermore, the study is anchored in a relatively brief panel dataset. Therefore, a longitudinal examination spanning extended durations would be instrumental in elucidating the evolution of migration dynamics in tandem with shifts in hydrological and broader environmental conditions.

## Methods
### Subnational migration
The data of population migration is ascertained by evaluating the annual fluctuations in population stock within each designated unit, after accounting for and excluding intrinsic population growth in the

respective cell[7,43,66]. Notably, the yearly population stock metrics are sourced from WorldPop, a reputable repository renowned for furnishing precise dynamic population distribution data at a granular resolution of 3 arc seconds (equivalent to ~100 m at the equator)[67].

In a bid to augment the precision of this stock population data, the study undertook adjustments to the subnational statistical population figures provided by WorldPop. This recalibration ensures that the data aligns seamlessly with the official population estimates articulated by the Population Division of the Department of Economic and Social Affairs within the United Nations Secretariat at a country-specific granularity. Furthermore, for the purposes of this study, we operated under the assumption that intrinsic population growth exhibits consistency either at the national or regional level. This postulation facilitates the computation of natural population growth for each cell, leveraging mortality and birth rate data procured from the World Bank. Finally, the variance in annual population stock, once the natural increase was discounted, is designated as the quantum of subnational population migration.

In addition, the study attempts to explore globally universal patterns of subnational population migration, but the pronounced population disparities across countries or regions present challenges to standardization. Therefore, the study characterized worldwide subnational migration by calculating Mig_R as the ratio of unit migration to the total population. In the meantime, the research calculated the Mig_R for minor, adult, elder, male, and female groups based on the above approach. It is noteworthy, however, that biases inherent in data estimation can occasionally lead to pronounced migration fluctuations in units with smaller volumes. To mitigate the potential skewness introduced by such outliers, the study implemented a 0.1% Winsorization, ensuring the integrity and robustness of the model outcomes[68].

### Hydrological intrusion exposure

Climate change is amplifying spatial and temporal alterations in global surface water. The historical repositories of Earth observation satellite data stand as the robust mechanism to elucidate these hydrological shifts at a global magnitude[69]. In this context, Pickens et al. constructed an unbiased estimate of the extent of dynamic surface water at a granular resolution of 30 m at the equator from 1999 to 2021 based on Landsat remotely sensed data[70]. It is noteworthy that hydrological datasets estimated in 2015 are punctuated with considerable data voids, attributable to constraints in the fidelity of the remote sensing satellite data available during that epoch. Consequently, the analytical purview is concentrated on hydrological intrusion between 2015 and 2020.

According to the IPCC, climate risk results from the interactions of exposure, hazard, and vulnerability[28]. This study employed a population-weighted hydrologic exposure index to quantify HIE. To match the resolution of the population data used, this paper used a mapping software called ArcGIS Pro 2.5.0, whereby the surface water raster data is transformed from an initial spatial resolution of 30 m to a resolution of 100 m. There were raster areas in the water dataset that do not contain any information (so-called null domains), which could have produced errors in the interpretation of the percentage of the population that was really affected by the proximity to water. To fix this, the areas in question were annotated as "null" in the population dataset, ensuring that the final analysis would not underestimate the true impact of water exposure on the population. The methodology to compute the population-weighted index is delineated as follows:

$$HIE_j = \frac{\sum_{i=1}^{n} Pop_i \times SurfaceWater_i}{\sum_{i=1}^{n} Pop_i} \quad (1)$$

Where $Pop_i$ represents the population of the $i$th raster unit; $SurfaceWater_i$ represents the proportion of surface water occurrence

during the year of the $i$th raster unit; and $n$ is the number of the raster unit within the subnational administrative unit.

$$HIE\_D_j = \frac{\frac{\sum_{i=1}^{m} HIE_i}{m} - HIE_j}{HIE_j} \quad (2)$$

Where $m$ is the number of neighborhood administrative units; $HIE_i$ represents the value of HIE of the $i$th neighborhood administrative unit; $HIE_j$ represents the value of HIE of the subnational administrative unit itself.

### Hydrological hazard

Hydrological hazards refer to natural events that can potentially cause harm and damages, such as the low frequency or high intensity rains and floods. However, due to certain biasness in global disaster databases, especially those recorded in underdeveloped regions, and inadequate coverage of smaller-scale disasters, this paper utilized the Max 5-day precipitation (Rx5day) as an indirect variable to represent hydrological hazards. Rx5day measures the highest total rainfall over a time-window of 5 consecutive days in a year, providing a globally consistent observation of the intensity of extreme events that could cause floods or other hydrological disasters. The basic rainfall datasets were extracted from the ERA5-Land dataset, which contains reliable and comprehensive coverage of hydrological events for establishing hydrological risk[71].

### Vulnerability

Vulnerability is conceptualized as an inherent susceptibility to detrimental impacts, a notion that is widely recognized to exhibit variations both intra-community and inter-society while also evolving temporally[28]. This paper performed a comprehensive assessment of vulnerability at the subnational administrative level, and included variables such as temperature, precipitation, elevation, and proximate distance from water bodies as environmental and geographical vulnerability. Leveraging on the toolbox of ArcGIS Pro 2.5.0, the writers were able to establish the regional mean values of these hazard determinants, integrating them as covariates within the empirical modeling framework. Pertinently, variables like TEMP and PRCP were inferred from the ERA5-Land dataset, a repository recognized for its consistent raster data characterized by both high spatial and temporal resolution, thereby fortifying hydrological research endeavors[71]. In parallel, the digital elevation model (DEM) was ascertained by leveraging the mapping resources furnished by the shuttle radar topography Mission (SRTM)[72]. Concerning Water_Dis, the approach entailed computing the mean distance from aquatic bodies for each designated grid.

$$Water\_Dis_j = \frac{\sum_{i=1}^{n} Distance_i}{n} \quad (3)$$

Where $Distance_i$ represents the distance from the water of the $i$th raster unit; and $n$ is the number of the raster unit within the subnational administrative unit.

Moreover, the study endeavors to gauge population vulnerability, delineating it through parameters like prosperity, economic standing, urbanization, and educational attainment. The prosperity metric, denoted as NTL_PA, was derived from nighttime light satellite imageries, which are calibrated to each other via the Defence Meteorological Satellite Program Operational Linescan System (DMSP-OLS) and Suomi National Polar-orbiting Partnership Visible Infrared Imaging Radiometer Suite (NPP-VIIRS)[73]. To perform consistent estimates of the level of the subnational scale economy worldwide, we calculated the GDP_PA variable by using GDP raster data based on top-down estimation[74]. The urbanization, symbolized as Urban_R, represents the

proportion of the urban populace relative to the aggregate demographic, with the urban contingent being discerned from WorldPop, utilizing land cover classification maps from the European Space Agency to define urban boundary[75]. Additionally, the educational benchmark, represented as Edu, was extrapolated from the mean years of formal education, sourced from the Subnational Human Development Index (SHDI)[76].

### Empirical model

With due consideration of the anticipated nonlinearity inherent in the relationship between Mig_R and hydrological intrusion risk variables, an ML model emerges as an appropriate approach to deal with the nuances of hydrological intrusion risks. The LightGBM algorithm, rooted in the gradient-boosting framework, stands as a reputable tool for predictive modeling within the ML discipline[77]. The LightGBM model includes three key enhancements that makes it an optimal choice for hydrological risk investigation[78]. Firstly, the gradient-based one-side sampling technique utilizes gradient information to efficiently sample data, prioritizing samples with high gradients while reducing the impact of those with low gradients. This approach places a focus on the most informative data points. Secondly, the Exclusive Feature Bundling method used tackles the issues of managing sparse and high-dimensional features by combining similar features, thereby reducing the complexity of the data without sacrificing essential information. Lastly, the histogram technique used enhances the generalization capabilities of the model by converting and splitting continuous data splits into discrete categories, simplifying data representation, and improving the efficiency of data processing. These techniques and advancements position LightGBM as a superior tool for developing a complex de-coupled model that aptly addresses the nuanced requirements of this study.

It is a well-known fact that while they offer certain precision, some high-accuracy nonlinear models often grapple with the challenge of an excessive number of parameters, which can be difficult to interpreted[79]. This study overcomes this challenge by incorporating the SHAP explanation algorithm, facilitating a transition from micro model explanations to a more macro or holistic understanding[80]. In this study, 80% of the dataset are allocated for training the model, reserving the remaining 20% for validation of the model. The performance of the model was assessed rigorously using a fivefold cross-validation technique.

Given that the characteristics of residence inherently influence residential migration and that migration is a temporal process, we introduced a temporal lag of one year to Mig_R. This adjustment ensures Mig_R can match the residence characteristics of the current year. Moreover, migration is a protracted behavior, implying that temporal trends could significantly influence the determinants of migration and their subsequent effects[3]. However, as this research is anchored in short-term panel data, we incorporated a two-year lag for Mig_R (denoted as MigR_Lag) to capture this migration trajectory. On the other hand, regional attributes such as cultural nuances, religious affiliations, and policy frameworks present challenges in achieving a consistent and effective global quantification. To address this, we integrated country-year fixed effects into the model, thereby providing a safeguard against potential distortions in the model outcomes attributable to omitted variables.

### Reporting summary

Further information on research design is available in the Nature Portfolio Reporting Summary linked to this article.

### Data availability

The data used in this study are available in the WorldPop database [https://hub.worldpop.org/]; Global surface water dynamics [https://glad. umd.edu/dataset/global-surface-water-dynamics]; ERA5-Land [https://www.ecmwf.int/en/forecasts/dataset/ecmwf-reanalysis-v5]; SRTM; NTL [https://dataverse.harvard.edu/dataset.xhtml?persistentId=doi:10.7910/DVN/YGIVCD]; GDP [https://doi.org/10.1038/s41597-022-01322-5]; SHDI [https://globaldatalab.org/shdi/] (Supplementary Table. 3).

### Code availability

Code for the main modeling and analysis process is available on Zenodo at https://doi.org/10.5281/zenodo.10911389.

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

## Acknowledgements

The study is financed by "Basic Theory of Sustainable Urban Planning, Construction, and Governance" under the 14th Five-Year Plan of the State Key Research and Development Program of the People's Republic of China (Grant No.2022YFC3800205, Z.W.) and 'Key Technologies for Regional Carbon Neutral Mega-City Planning and Design' under Shanghai Science and Technology Support Program for Carbon (Grant No.22DZ1207800, Z.W.).

## Author contributions

R.Q.: Conceptualization, methodology, formal analysis, writing—original draft, project administration. S.G.: Conceptualization, writing—original draft. X.L.: Writing—review & editing, resources. L.X.: Writing—original draft, data curation, G.Z.: data curation, visualization. X.M.: Formal analysis, Writing—original draft. Z.L.: Conceptualization, data curation, formal analysis, M.W.: Conceptualization, writing—review & editing. S.Z.: Conceptualization, writing—original draft. Z.W.: Conceptualization, writing—review & editing, supervision.

## Competing interests
The authors declare no competing interests.
