## [Peer Review File · Nature Communications]

From Exodus to Trapping Understanding the Global Subnational Migration Patterns Driven by Hydrological Intrusion ExposureReviewers' Comments:

Reviewer #1:

Remarks to the Author:

This paper presents an interesting approach to elucidate the phenomenon of migration in a micro-scale. Hydrologic intrusion exposure was used as the main driving parameter for migration. Various others socio-economic and physical parameters were also considered.

While the findings of the study would be of interest to the readers, the paper as it is, is not suready for publications. The presentation, both in terms of logical development and use of English need substantial revision. In particular, the narrative to use HIE and non-linear model is difficult to comprehend. The readers would also need to be fairly competent in statistics and certain socio-economic knowledge to fully appreciate the findings of the study. The discussion presented came through as statements rather than presenting the findings, justifications and drawing observations. Not withstanding these flaws, the paper, once revised with the help of a competent English speaker, would be a valuable contribution to the journal.

This paper also came through as the written work of several writers with varying competence level in English writing. The paper would benefit from a harmonised presentation. The presentation also dealing with numbers referring to small percentages. The authors need to elaborate on the significance of these small percentages, variant and the reasons behind making presentation in these manners.

The figures, while interesting, need to improve on the presentation, particularly the size of the figures, legends, font-size of the axis title, etc.

This reviewer would like to recommend that the authors make a substantial revision of the paper and resubmit after revision as this is a potentially paper of interest a

Reviewer #2:

Remarks to the Author:

The study assesses the hydrological risk on global subnational migration over 46,000 global subnational units, utilizing remote sensing to measure hydrological intrusion exposure. The results highlight the importance of HIE in migration, surpassing socioeconomic and hazard factors. The topic is timely in the context of climate change. While the results are clear, the justification and explanation of the methodology should be expanded.

My primary concern is the risk assessment. While the definitions of hazard, exposure, and vulnerability are clearly outlined in P4, I find a discrepancy in the selected hazard indicators. Specifically, variables such as temperature, precipitation, elevation, and proximate distance from water bodies do not align with their respective hazard definitions. Generally, hazards pertain to flood-related risks like the frequency or intensity of floods and storms. Temperature, on its own, does not pose a hazard; instead, the associated risks should be identified as extreme heat or heat waves. Moreover, factors like elevation and proximate distance from water bodies should be categorized under ecological, environmental, or geographical vulnerability rather than hazard.

The variables lack clarity, particularly regarding each risk component. It is unclear which variables correspond to each hazard, and the rationale behind their selection for risk assessment is not clearly explained. To enhance understanding, I recommend incorporating a table that comprehensively explains all hazard, exposure, and vulnerability variables.

Some potential readers may find it challenging to comprehend the methodology, especially in the context of AI training. To enhance understanding, I recommend providing additional explanations or

incorporating images to illustrate the calculation process. This could be included within the method section or as supplementary materials.

Another aspect of concern revolves around the implications of this study. While the results underscore the significant role of hydrological risk in migration, it is crucial to highlight how the research outcomes can be utilized to support or intervene in mitigation policies. Emphasizing the practical application of the study can enhance its overall importance.

Further clarification is needed on data availability. It is essential to explicitly outline the data sources in the Data Availability section to facilitate easy data retrieval and queries.

Consider incorporating supplementary material, such as a list of countries, to enhance the comprehensiveness of the study.

Please align with the journal format, such as removing or merging the Conclusion section.

Responses to Reviewer #1's comments:

This paper presents an interesting approach to elucidate the phenomenon of migration in a micro-scale. Hydrologic intrusion exposure was used as the main driving parameter for migration. Various others socio-economic and physical parameters were also considered.

Response:

Thank you for highlighting the unique aspects of our approach to investigate migration phenomena at a micro-scale. We appreciate your taking the time to review our work and are open to any suggestions or comments that could further enhance our work and the quality of the paper.

Comment 1:

While the findings of the study would be of interest to the readers, the paper as it is, is not ready for publications. The presentation, both in terms of logical development and use of English need substantial revision. In particular, the narrative to use HIE and nonlinear model is difficult to comprehend. The readers would also need to be fairly competent in statistics and certain socio-economic knowledge to fully appreciate the findings of the study. The discussion presented came through as statements rather than presenting the findings, justifications and drawing observations. Notwithstanding these flaws, the paper, once revised with the help of a competent English speaker, would be a valuable contribution to the journal.

Response:

Thank you for your feedback on the submitted manuscript, the presentation, logical development, and language use. The concerns regarding the narrative on HIE and the nonlinear model, as well as the readers' appreciation on statistics and socio-economics were duly noted.

In revising the paper, we have sought advice from a professional editor who is proficient in both English and our field of study. The revised manuscript now includes a more

detailed explanation of HIE and the nonlinear modeling approach. We strive to present the work in a simpler manner so that the readers could understand them and appreciate the significant scientific details. Furthermore, to make our paper more accessible to all readers, we try, as much as possible, not including specialized terms used in statistics or economics, but opting for more layman terms and expressions.

Following your guidance, we revised the discussion section of our paper to introduce the research findings, explain the rationale, and present our insights in a way that enhances the analysis and interpretation of our results. Moreover, we endeavor to relate our findings with the current state of the arts, fill in the knowledge gaps, thereby elevating the significance of our conclusions.

Corresponding modified content:

HIE Enhances Population Emigration

The results of the global model validated the conclusions of reported by others, and reaffirm that increasing hydrological risks lead to regional population displacement. Furthermore, the results also revealed a subtle trend: the migration effect diminishes as the risk exposure increases (as illustrated in Fig. 2_b). Specifically, when the HIE falls below 6.74%, an increase in HIE by one standard deviation (1.51%) corresponds to an average increase in the population out-migration rate of 0.09% (with a 95% confidence interval ranging from -0.11 % to -0.06%, according to the regression curve). Notably, a higher HIE (> 6.74%) appears not favoring outward migration, as is reflected clearly by the inflection point in the regression curve. This is plausibly due to hydrological risks which resulted in regional resource scarcity, which in turn limited the ability of the inhabitants to migrate.

This study sheds light on an unexplored phenomenon. Upon comparing the extent of exposure to hydrological risks in a region with that of its neighbors, one may detect the difference (which we term HIE_D) which significantly amplifies the likelihood of people moving away. Different from HIE, HIE_D reveals an enhanced impact (as shown in Figure 2_b), indicating that certain groups of individuals become more sensitive to HIE when there exists other options and alternatives. In more tangible terms, as the disparity in HIE exceeds 0%, an increase in HIE by one standard deviation (127.42%) corresponds to a rise in Mig_R by 0.34% (95% CI -0.36% ~ -0.33%). Simply put, when people perceive their area to be at a higher risk compared

to that in neighboring regions, the inclination to migrate intensifies.

As for the middle-income group, the effects of HIE is similar to that of the aforementioned global modelling result, albeit with a lower suppression threshold (as illustrated in Fig. 3_d). As HIE falls below 5.00%, an increase in HIE by one standard deviation corresponds to an average decrease in Mig_R of 0.09% (95% CI -0.09% ~ -0.08%). Compared to the other two groups, the middle-income group appears insensitive to HIE_D. This could be attributed to the fact that, unlike the low-income group, the middle-income group has certain resources to relocate readily., Unlike the high-income group, the middle-income group may not have that much resources to weather through more vulnerability, and relocation is then a necessary option.

When it comes to sex disparities, the findings of this study show that environmental factors affect both sexes in similar ways, but the effects are more noticeable in men. Specifically, the significance of HIE and HIE_D in influencing decisions is marginally higher for men than women, at 14.82% and 13.04%, respectively. This suggests that men might be more responsive to negative changes in their environment, and migrate particularly when migration often leads to improved earnings. On the other hand, for women, the link between migration and earnings appears not as strong, and are therefore less receptive to relocate⁵⁴.

Hydrological Intrusion Exposure is not a Weak Variable

From the results of the global and subgroup models, one can see that the impact of HIE and HIE_D on migration rates often registers below 10%—indicating a seemingly minor role. Similarly, some reported studies have also shown that climatic conditions are weak predictors of migration^{20,55}. Nonetheless, one does not dismiss the significance of hydrological factors on population movements as inconsequential. Particularly notable is the role of HIE_D: when the HID-value exceeds the mean value, a 1.35% decrease in Mig_R and represented the most significant reduction attributed to all other parameters examined. The caveat here is the small number of samples that exceed the mean, which, in turn, appear to marginalize the overall importance of HIE_D in the broader analysis. This finding underscores the resilience of a population group to risks, suggesting that while hydrological variables may not always be the primary drivers, their impact, especially in specific contexts like that of HIE_D-value exceeding the mean value, can be substantial.

Among the various identified drivers of migration, the economic condition of a

region stands out at a relative importance of 13.13% in driving population shifts. However, the influence of economic condition on Mig_R predominantly hovers around the national average, with a Mig_R improvement of 0.40% within one standard deviation in economy (6.75 million dollars per square kilometer) above and below the mean. This suggests that while immediate economic enhancements may effectively curb population decline, achieving higher than the average national economy does not necessarily hold the residents more firmly to the home region. In contrast, mitigating hydrological risks emerges as a more viable, long-term strategy to deter out-migration.

Earlier reported studies might have overlooked this nuance, and might have simplistically mapped environmental factors to migration in a linear fashion. The intervention policy might then be less than optimal. Based on the findings of this study, the writers advocate for a refined perspective, recognizing that while economic development is crucial, environmental risk mitigation efforts, especially mitigating the differences and gaps with neighboring regions, are positive long-term measures that could be used to manage population displacement more effectively.

Vulnerability and Decision to Migrate

Economic development is pivotal in orchestrating the influence dynamics of hydrological encroachment on cross-border migration patterns⁶¹. This role is clearly elucidated in this study and is discernible at the sub-national level. Notably, a high-income region portrays remarkable resilience, wherein even locales fraught with heightened hydrological jeopardy ($HIE \geq 2.40\%$) exert a pull on inhabitants due to their strong economic status. In contrast, regions characterized by low-income strata show a tendency to stay put in the face of risks, wherein the parameter denoting HIE_D (relative importance at 10.52%) ascends to prominence as the most impactful parameter with respect to HIE and hazardous conditions. This conveys a double whammy confronting economically disadvantaged groups, i.e. increased vulnerability to calamities and dwindling resources to escape from them. As a result, migration, if at all is restricted to adjoining areas.

This observation is reflective of those certain demographic segments, notably older adults and women, who are exposed to heightened vulnerability, and are ensnared in regions with extensive hydrological risk. An added observation is that the elderly demographic manifests pronounced tendencies towards a stable environment and the

desire to move away from metropolitan areas. Such trends can be ascribed to an amalgamation of reasons, such as the metaphorical "falling leaves returning to the roots (getting home)" cultural concept and a passive stance due to aging and weaker physical conditions⁶². On the other hand, the characteristics of the population group play a vital role in shaping migration patterns. Objective physical factors such as employment opportunity, and subjective perceptions of risk, cultural norms, and religious beliefs can significantly moderate decision-making processes, either facilitating or resisting migration^{63,64}.

Hydrological Effects Include Various Heterogeneous Threshold Levels

Migration may be viewed as both a response and pre-emptive measure to climate change, and constitute certain potential strategies for resettlement⁶⁵. This study reveals that the impact of environmental changes on migration is nonlinear, and is especially evident in the exposure parameter HIE_D and the hazard parameter Rx5day. Both of these parameters demonstrate the resilience of the population group against environmental stresses, with out-migration occurring only after the critical threshold is exceeded (-37.15% for HIE_D and 135.66mm for Rx5day). This may be the direct consequence of the population group being overwhelmed, ceased to accept and unable to adapt to emerging pressures^{66,67}. The identification of these thresholds is crucial, as the threshold signifies the limits of adaptability and resilience within the population group, beyond which significant population movements would be observed.

Hydrological Intrusion Exposure

According to the IPCC, climate risk results from the interactions of exposure, hazard, and vulnerability²⁸. This study employed a population-weighted hydrologic exposure index to quantify HIE. To match the resolution of the population data used, this paper used a mapping software called ArcGIS Pro 2.5.0 whereby the surface water raster data is transform from an initial spatial resolution of 30 meters to a resolution of 100 meters. There were raster areas in the water dataset that do not contain any information (so called "null domains"), which could have produced errors in the interpretation of the percentage of population that was really affected by the proximity to water To fix this, the areas In questions were annotated as 'null' in the population dataset, ensuring that the final analysis would not underestimate the true impact of water exposure on the population.

Empirical Model

With due consideration on the anticipated nonlinearity inherent in the relationship between Mig_R and hydrological intrusion risk variables, an ML model emerges as an appropriate approach to deal with the nuances of hydrological intrusion risks. The LightGBM algorithm, rooted in the gradient-boosting framework, stands as a reputable tool for predictive modelling within the ML discipline⁷⁶. The LightGBM model includes three key enhancements that makes it an optimal choice for hydrological risk investigation⁷⁷. Firstly, the Gradient-based One-Side Sampling technique used utilizes gradient information to efficiently sample data, prioritizing samples with high gradients while reducing the impact of those with low gradients. This approach places a focus on the most informative data points. Secondly, the Exclusive Feature Bundling method used tackles the issues of managing sparse and high-dimensional features by combining similar features, thereby reducing the complexity of the data without sacrificing essential information. Lastly, the Histogram Technique used enhances the generalization capabilities of the model by converting and splitting continuous data splits into discrete categories, simplifying data representation, and improving the efficiency of data processing. These techniques and advancements position LightGBM as a superior tool for developing a complex de-coupled model that aptly addresses the nuanced requirements of this study.

It is a well-known fact that while they offer certain precision, some high-accuracy nonlinear models often grapple with the challenge of an excessive number of parameters, which can be difficult to interpreted. This study overcome this challenge by incorporating the SHAP explanation algorithm, facilitating a transition from micro model explanations to a more macro or holistic understanding⁷⁸. The computational framework for SHAP values is shown as follows⁷⁹:

$$SHAP_j = \sum_{S \subseteq [V_1 + V_2 + \dots + V_p] \setminus [V_j]} \frac{|S|! (p - |S| - 1)!}{p!} (f_x(S \cup [V_j]) - f_x(S))$$
$$y_i = y_{base} + \sum_{j=1}^k SHAP(x_{i,j})$$

Here, $SHAP_j$ represents the SHAP value corresponding to feature j ; S denotes the subset of features incorporated in the model; V_p is the feature of the model; p is the total count of features; $f_x(S)$ is the predicted value produced by the model for the

given subset; y_i refers to the predicted value for the respective i^{th} sample; y_{base} is the mean predicted value across all other samples; $SHAP(x_{i,j})$ is the SHAP value of the feature j at sample i ; and k is the number of features. In this study, 80% of the dataset are allocated for training the model, reserving the remaining 20% for validation of the model. The performance of the model was assessed rigorously using a 5-fold cross-validation technique.

Comment 2:

This paper also came through as the written work of several writers with varying competence level in English writing. The paper would benefit from a harmonised presentation. The presentation also dealing with numbers referring to small percentages. The authors need to elaborate on the significance of these small percentages, variant and the reasons behind making presentation in these manners.

Response:

Thank you for your guidance. We have reviewed the manuscript thoroughly to ensure consistency in style, tone, and terminology.

With respect to the small percentages referred to in the paper, it is noted that, these small numbers, accumulated over the entire duration of many years turn out to be significant. The corresponding impacts are amplified impact in densely populated areas. Even an apparent minor variation produces significant long-term and widespread effects on population migration. We standardized the retention of data to two decimal places or two significant digits in various places to improve readability of the paper.

Corresponding modified content:

Uneven Distribution of Subnational Population Migration

The average Mig_R value across global regions stands at -0.59%, signifying that a large portion of the world population is experiencing out-migration. Although the magnitude of Mig_R appears small, the implications are profound, especially when the number is applied to the global population of 8 billion. The World Migration Report 2022 reported a total of 281 million international migrants which is in no way insignificant. The cumulative effects of small changes or variations over time and

their intensified impacts in densely populated areas, exact significant long-term and widespread effects on migration patterns worldwide.

Comment 3:

The figures, while interesting, need to improve on the presentation, particularly the size of the figures, legends, font-size of the axis title, etc.

Response:

We have made significant adjustments to improve the clarity and readability of the figures. Specifically, we have increased the size of the figures, improved the legibility of the legends, and enlarged the font size of the axis titles, making it easier for readers to appreciate the data and analyses presented.

Corresponding modified figures and contents:

Fig. 1. Regional estimates of Mig_R and HIE

Fig. 2. The effect of per feature on Mig_R.

Fig. 3. The effect of per feature on Mig_R.

Fig. 4. The effect of per feature on Mig_R.

Responses to Reviewer #2's comments:

The study assesses the hydrological risk on global subnational migration over 46,000 global subnational units, utilizing remote sensing to measure hydrological intrusion exposure. The results highlight the importance of HIE in migration, surpassing socio-economic and hazard factors. The topic is timely in the context of climate change. While the results are clear, the justification and explanation of the methodology should be expanded.

Response:

Thank you for your concurrence on the significance of our study on hydrological risks and for acknowledging the timeliness of this work . We appreciate your feedback and suggestions on improving the presentation of our results. We concur and understand the need to provide a more comprehensive justification and explanation of our methodology. In response to your comments, we have expanded the section on the methodology used. We believe this revised paper will provide readers and fellow researchers with a clearer understanding of our approach, and the basis upon which our findings are based.

Comment 1:

My primary concern is the risk assessment. While the definitions of hazard, exposure, and vulnerability are clearly outlined in P4, I find a discrepancy in the selected hazard indicators. Specifically, variables such as temperature, precipitation, elevation, and proximate distance from water bodies do not align with their respective hazard definitions. Generally, hazards pertain to flood-related risks like the frequency or intensity of floods and storms. Temperature, on its own, does not pose a hazard; instead, the associated risks should be identified as extreme heat or heat waves. Moreover, factors like elevation and proximate distance from water bodies should be categorized under ecological, environmental, or geographical vulnerability rather than hazard.

Response:

Your feedback highlighted the importance of defining and characterizing hydrological hazards, a crucial element of our study. We understand that hydrological hazards typically refer to natural events that could be harmful or damaging, such as heavy rains and floods, as reflected by the frequency and intensity of such events. We noted that the global disaster databases show a certain degree of biasness in recording disasters and extreme events in underdeveloped regions, especially in capturing smaller-scale disasters, which could affect our assessment of hydrological hazards. Therefore, we chose Max 5-day precipitation (Rx5day) as a key variable to represent hydrological hazards. Rx5day measures the highest amount of precipitation over five consecutive days within a year, providing a globally consistent observation of the intensity of extreme events that could cause floods or other hydrological disasters. This method helps us better identify and quantify extreme hydrological events that could have significant socio-economic impacts.

We noted your advice duly, and have also adjusted other variables, such as elevation and proximity to water bodies, reclassifying them as factors reflecting regional vulnerability. This adjustment aims to provide more scientifically sound basis in mapping the sensitivity and vulnerability of different regions to hydrological hazards, thereby offering a more comprehensive and precise risk assessment framework.

We greatly appreciate your valuable suggestions, which help us improve our research methodology significantly and further our understanding of hydrological risk assessment. We strive to incorporate these adjustments in details in the revised version of our paper.

Corresponding modified content:

Hydrological Hazard

Hydrological hazards refer to natural events that can potentially cause harm and damages, such as the low frequency or high intensity rains and floods. However, due to certain biasness in global disaster databases, especially those recorded in underdeveloped regions, and inadequate coverage of smaller-scale disasters, this paper utilized the Max 5-day precipitation (Rx5day) as an indirect variable to represent hydrological hazards. Rx5day measures the highest total rainfall over a

time-window of five consecutive days in a year, providing a globally consistent observation of the intensity of extreme events that could cause floods or other hydrological disasters. The basic rainfall datasets were extracted from the ERA5-Land dataset, which contains reliable and comprehensive coverage of hydrological events for establishing hydrological risk⁷⁴.

Vulnerability

Vulnerability is conceptualized as an inherent susceptibility to detrimental impacts, a notion that is widely recognized to exhibit variations both intra-community and inter-society while also evolving temporally²⁸. This paper performed a comprehensive assessment of vulnerability at the subnational administrative level, and included variables such as temperature, precipitation, elevation, and proximate distance from water bodies as environmental and geographical vulnerability. Leveraging on the toolbox of ArcGIS Pro 2.5.0, the writers were able to establish the regional mean values of these hazard determinants, integrating them as covariates within the empirical modelling framework.

Comment 2:

The variables lack clarity, particularly regarding each risk component. It is unclear which variables correspond to each hazard, and the rationale behind their selection for risk assessment is not clearly explained. To enhance understanding, I recommend incorporating a table that comprehensively explains all hazard, exposure, and vulnerability variables.

Response:

Thank you for your valuable feedback about providing clear definition of each risk component in the manuscript. We have added a table in the revised paper. This table contains comprehensive details of all the variables associated with hazard, exposure, and vulnerability, and clearly indicates which variables correspond to which component. It also offers statistical details such as means and standard deviations, which aid users in appreciating the distribution and significance of the variables and in assessing hydrological risks.

Corresponding modified content:				
	Variable	Description	Mean	SD
Exposure Variable	HIE (%)	This metric assesses the region's exposure level to hydrological risk.	1.35	2.30
	HIE D (%)	This metric measures the difference in HIE between neighboring areas.	-36.25	148.72
Hazard Variable	Rx5day (mm)	Represents the highest precipitation received over five consecutive days within a year.	104.56	67.60
Vulnerability Variable	PRCP (mm)	General measure of rainfall, providing an overview of water availability and potential flood risk.	42.68	29.65
	Water_Dis (km)	The proximity of a location to rivers, lakes, or other water bodies.	87.92	101.09
	TEMP (°C)	Average temperature measurement.	16.66	8.20
	DEM (m)	A representation of a region's elevation.	388.53	545.16
	GDP_PA (Million dollars/km ²)	Economic productivity is measured, indicating regional economic strength and development.	6.06	15.18
	NTL_PA	Satellite imagery measuring artificial lighting at night.	2.31	8.60
	Urban_R (%)	The rate of urbanization.	17.16	22.58
	Edu (Years)	Average years of education of the population.	9.95	3.02
Supplementary Table. 1 The variables used in defining hydrological risk Noted: This study employed a dataset that consists of eleven categories of variables and they were used to describe the hydrological risk in exposure, hazard, and vulnerability.				

Comment 3:

Some potential readers may find it challenging to comprehend the methodology, especially in the context of AI training. To enhance understanding, I recommend providing additional explanations or incorporating images to illustrate the calculation process. This could be included within the method section or as supplementary materials.

Response:

In response to your feedback, we have added further elaboration in the methodology section, and included more detailed explanations of the training process of the artificial

intelligence (AI) system used. Several relevant formulas are also added. This enhancement aims to provide readers with a clearer understanding of the AI concepts and computational procedures we employed, ensuring that the intricacies of our approach are clear and readily comprehensible.

Corresponding modified content:

Empirical Model

With due consideration on the anticipated nonlinearity inherent in the relationship between Mig_R and hydrological intrusion risk variables, an ML model emerges as an appropriate approach to deal with the nuances of hydrological intrusion risks. The LightGBM algorithm, rooted in the gradient-boosting framework, stands as a reputable tool for predictive modelling within the ML discipline⁷⁶. The LightGBM model includes three key enhancements that makes it an optimal choice for hydrological risk investigation⁷⁷. Firstly, the Gradient-based One-Side Sampling technique used utilizes gradient information to efficiently sample data, prioritizing samples with high gradients while reducing the impact of those with low gradients. This approach places a focus on the most informative data points. Secondly, the Exclusive Feature Bundling method used tackles the issues of managing sparse and high-dimensional features by combining similar features, thereby reducing the complexity of the data without sacrificing essential information. Lastly, the Histogram Technique used enhances the generalization capabilities of the model by converting and splitting continuous data splits into discrete categories, simplifying data representation, and improving the efficiency of data processing. These techniques and advancements position LightGBM as a superior tool for developing a complex de-coupled model that aptly addresses the nuanced requirements of this study.

It is a well-known fact that while they offer certain precision, some high-accuracy nonlinear models often grapple with the challenge of an excessive number of parameters, which can be difficult to interpreted. This study overcome this challenge by incorporating the SHAP explanation algorithm, facilitating a transition from micro model explanations to a more macro or holistic understanding⁷⁸. The computational framework for SHAP values is shown as follows⁷⁹:

$$SHAP_j = \sum_{S \subseteq [V_1+V_2+\dots+V_p] \setminus [V_j]} \frac{|S|! (p - |S| - 1)!}{p!} (f_x(S \cup [V_j]) - f_x(S))$$

$$y_i = y_{base} + \sum_{j=1}^k SHAP(x_{i_j})$$

Here, $SHAP_j$ represents the SHAP value corresponding to feature j ; S denotes the subset of features incorporated in the model; V_p is the feature of the model; p is the total count of features; $f_x(S)$ is the predicted value produced by the model for the given subset; y_i refers to the predicted value for the respective i^{th} sample; y_{base} is the mean predicted value across all other samples; $SHAP(x_{i_j})$ is the SHAP value of the feature j at sample i ; and k is the number of features. In this study, 80% of the dataset are allocated for training the model, reserving the remaining 20% for validation of the model. The performance of the model was assessed rigorously using a 5-fold cross-validation technique.

Comment 4:

Another aspect of concern revolves around the implications of this study. While the results underscore the significant role of hydrological risk in migration, it is crucial to highlight how the research outcomes can be utilized to support or intervene in mitigation policies. Emphasizing the practical application of the study can enhance its overall importance.

Response:

Thank you for your insightful feedback regarding the implications of our study. We acknowledge the important role of policy and policy making. We have not only expanded our discussion on the impact of hydrological risks on migration but also included a real-world example of the Libyan floods as an illustration. In addition, we included various suggestions on viable policies in the discussion section, ensuring a coherent narrative that underscores practical relevance and applications of our findings.

Corresponding modified content:

Risk Reduction Policy

The work render research gains added relevance in light of recent extreme hydrological events such as the Libyan floods. The extreme precipitation in Libya swiftly overwhelmed local resilience, stranding tens of thousands in flooded areas.

There is an imperative need for enhanced early warning systems and community awareness programs. These should be designed to not only alert communities of impending dangers but also educate and empowering them on effective response strategies. Strengthening infrastructure resilience through the reinforcement of flood defenses and development of appropriate sustainable water management solutions are also crucial. This catastrophic event is a stark reminder of the increased hydrological risks that vulnerable communities increasingly face. It also highlights the urgent need for comprehensive, and adaptive strategies in addition to immediate risk mitigation, including long-term preparedness for future hydrological intrusion events.

To mitigate the impact of natural disasters, it is crucial to enhance infrastructure in economically disadvantaged areas. This involves strengthening housing structures, constructing flood defenses, and upgrading drainage systems to offer better protection to residents during such events, thereby diminishing the necessity for relocation. Furthermore, the provision of comprehensive social support and services is essential. This includes access to healthcare, services catering to the elderly, and economic assistance tailored for individuals with limited physical abilities or those belonging to other vulnerable groups. Implementing these measures aims to alleviate the burden of natural risks, enhancing the capability of individuals and communities to manage and respond to natural disasters effectively.

Comment 5:

Further clarification is needed on data availability. It is essential to explicitly outline the data sources in the Data Availability section to facilitate easy data retrieval and queries.

Response:

In response to the request for further clarification on data availability, we have updated the Data Availability section of our manuscript to provide a comprehensive and explicit outline of all data sources used in our study.

Corresponding modification content:

Data Availability

Data relevant to this study can be downloaded from the following website (Supplementary Table. 3): WorldPop, <https://hub.worldpop.org/>; Global surface water dynamics, <https://glad.umd.edu/dataset/global-surface-water-dynamics>; ERA5-Land, <https://www.ecmwf.int/en/forecasts/dataset/ecmwf-reanalysis-v5>; SRTM, <https://www.earthdata.nasa.gov/sensors/srtm>; NTL, <https://dataverse.harvard.edu/dataset.xhtml?persistentId=doi:10.7910/DVN/YGIVCD>; GDP, <https://doi.org/10.1038/s41597-022-01322-5>; SHDI, <https://globaldatalab.org/shdi/>.

Data	Source
WorldPop	https://hub.worldpop.org/
Global surface water dynamics	https://glad.umd.edu/dataset/global-surface-water-dynamics
ERA5-Land	https://www.ecmwf.int/en/forecasts/dataset/ecmwf-reanalysis-v5
SRTM	https://www.earthdata.nasa.gov/sensors/srtm
NTL	https://dataverse.harvard.edu/dataset.xhtml?persistentId=doi:10.7910/DVN/YGIVCD
GDP	https://doi.org/10.1038/s41597-022-01322-5
SHDI	https://globaldatalab.org/shdi/

Supplementary Table. 3 Data sources

We have also included supplementary material featuring a detailed list of countries that were analyzed. This added details is aimed to bolster transparency, facilitate further research, and assist policymakers by clearly delineating the geographical scope of our analysis.

Corresponding modification content:	
Region	ISO
Australia/New Zealand	AUS; CCK; CXR; HMD; NFK; NZL
Caribbean	ABW; AIA; ATG; BES; BHS; BLM; BRB; CUB; CUW; CYM; DMA; DOM; GLP; GRD; HTI; JAM; KNA; LCA; MAF; MSR; MTQ; PRI; SXM; TCA; TTO; VCT; VGB; VIR
Central America	BLZ; CRI; GTM; HND; MEX; NIC; PAN; SLV; XCL
Central Asia	KAZ; KGZ; TJK; TKM; UZB
Eastern Africa	BDI; COM; DJI; ERI; ETH; KEN; MDG; MOZ; MUS; MWI; MYT; REU; RWA; SOM; SSD; SYC; TZA; UGA; ZMB; ZWE
Eastern Asia	CHN; HKG; JPN; KOR; MAC; MNG; PRK; TWN; XPI; XSP
Eastern Europe	BGR; BLR; CZE; HUN; MDA; POL; ROU; RUS; SVK; UKR

Melanesia	FJI; NCL; PNG; SLB; VUT
Micronesia	FSM; GUM; KIR; MHL; MNP; NRU; PLW
Middle Africa	AGO; CAF; CMR; COD; COG; GAB; GNQ; STP; TCD
Northern Africa	DZA; EGY; ESH; LBY; MAR; SDN; TUN
Northern America	BMU; CAN; GRL; SPM; UMI; USA
Northern Europe	ALA; DNK; EST; FIN; FRO; GBR; GGY; IMN; IOT; IRL; ISL; JEY; LTU; LVA; NOR; SGS; SJM; SWE; BVT; PCN; XAD
Polynesia	ASM; COK; NIU; PYF; TKL; TON; TUV; WLF; WSM
South America	ARG; BOL; BRA; CHL; COL; ECU; FLK; GUF; GUY; PER; PRY; SUR; URY; VEN
South-Eastern Asia	BRN; IDN; KHM; LAO; MMR; MYS; PHL; SGP; THA; TLS; VNM
Southern Africa	BWA; LSO; NAM; SWZ; ZAF
Southern Asia	AFG; BGD; BTN; IND; IRN; LKA; MDV; NPL; PAK
Southern Europe	ALB; AND; BIH; ESP; GIB; GRC; HRV; ITA; MKD; MLT; MNE; PRT; SMR; SRB; SVN; VAT; XKO
Western Africa	BEN; BFA; CIV; CPV; GHA; GIN; GMB; GNB; LBR; MLI; MRT; NER; NGA; SEN; SHN; SLE; TGO
Western Asia	ARE; ARM; AZE; BHR; CYP; GEO; IRQ; ISR; JOR; KWT; LBN; OMN; PSE; QAT; SAU; SYR; TUR; YEM; XNC
Western Europe	ATF; AUT; BEL; CHE; DEU; FRA; LIE; LUX; MCO; NLD
Supplementary Table. 2 The list of countries	

Comment 6:

Please align with the journal format, such as removing or merging the Conclusion section.

Response:

We have integrated the content of the Conclusion section into the Discussion part of our manuscript. This approach maintains the integrity of our findings while adhering to the publication's structural preferences.

Reviewers' Comments:

Reviewer #1:

Remarks to the Author:

Thank you very much for the effort to revise the manuscript and respond to my suggestions. I am happy with the response and revision.

Please do a careful editorial and proof-read for the final submission

Reviewer #2:

Remarks to the Author:

My concerns have been addressed in the revision. Just one more minor comment. In Supplementary Table. 1, references are encouraged to support the rationale behind variable selection.

Responses to Reviewer #1's comments:

Thank you very much for the effort to revise the manuscript and respond to my suggestions. I am happy with the response and revision.

Response:

Thank you very much for your constructive feedback throughout the revision process. We are pleased that you are happy with our responses and the revisions made to the manuscript.

Responses to Reviewer #2's comments:

My concerns have been addressed in the revision. Just one more minor comment. In Supplementary Table. 1, references are encouraged to support the rationale behind variable selection.

Response:

We appreciate your positive response to our revisions and are glad to hear that our manuscript meets your expectations.

We appreciate your additional comment regarding Supplementary Table 1. We will incorporate references to support the rationale behind variable selection and update the supplementary materials accordingly.

Corresponding modified content:					
	Variable	Description	Mean	SD	Ref.
Exposure Variable	HIE (%)	This metric assesses the region's exposure level to hydrological risk.	1.35	2.30	
	HIE_D (%)	This metric measures the difference in HIE between neighboring areas.	-36.25	148.72	
Hazard Variable	Rx5day (mm)	Represents the highest precipitation received over five consecutive days within a year.	104.56	67.60	¹
Vulnerability Variable	PRCP (mm)	General measure of rainfall, providing an overview of water availability and potential flood risk.	42.68	29.65	²
	Water_Dis (km)	The proximity of a location to rivers, lakes, or other water bodies.	87.92	101.09	²
	TEMP (°C)	Average temperature measurement.	16.66	8.20	³
	DEM (m)	A representation of a region's elevation.	388.53	545.16	²
	GDP_PA (Million dollars/km ²)	Economic productivity is measured, indicating regional economic strength and development.	6.06	15.18	⁴

NTL_PA	Satellite imagery measuring artificial lighting at night.	2.31	8.60	⁵
Urban_R (%)	The rate of urbanization.	17.16	22.58	⁵
Edu (Years)	Average years of education of the population.	9.95	3.02	³

Supplementary Table. 1 The variables used in defining hydrological risk

Noted: This study employed a dataset that consists of eleven categories of variables and they were used to describe the hydrological risk in exposure, hazard, and vulnerability.

- 1 Zhang, W., Zhou, T., Zou, L., Zhang, L. & Chen, X. Reduced exposure to extreme precipitation from 0.5 °C less warming in global land monsoon regions. *Nature Communications* **9**, 3153 (2018). <https://doi.org/10.1038/s41467-018-05633-3>
- 2 Hauer, M. E. *et al.* Sea-level rise and human migration. *Nature Reviews Earth & Environment* **1**, 28-39 (2020). <https://doi.org/10.1038/s43017-019-0002-9>
- 3 Brottrager, M., Crespo Cuaresma, J., Kniveton, D. & Ali, S. H. Natural resources modulate the nexus between environmental shocks and human mobility. *Nature Communications* **14**, 1393 (2023). <https://doi.org/10.1038/s41467-023-37074-y>
- 4 Schutte, S., Vestby, J., Carling, J. & Buhaug, H. Climatic conditions are weak predictors of asylum migration. *Nature Communications* **12**, 2067 (2021). <https://doi.org/10.1038/s41467-021-22255-4>
- 5 Niva, V. *et al.* World's human migration patterns in 2000–2019 unveiled by high-resolution data. *Nature Human Behaviour* **7**, 2023-2037 (2023). <https://doi.org/10.1038/s41562-023-01689-4>